# Analysis of Diffusion Characteristics and Influencing Factors of Particulate Matter in Ship Exhaust Plume in Arctic Environment Based on CFD

**Yuanqing Zhu *** , **Qiqi Wan, Qichen Hou, Yongming Feng** , **Jia Yu, Jie Shi and Chong Xia**

College of Power and Energy Engineering, Harbin Engineering University, Harbin 150001, China;
wanqiqi@hrbeu.edu.cn (Q.W.); houqichen@hrbeu.edu.cn (Q.H.); fengyongming@hrbeu.edu.cn (Y.F.);
yuz@hrbeu.edu.cn (J.Y.); shi@hrbeu.edu.cn (J.S.); xiachong@hrbeu.edu.cn (C.X.)
* Correspondence: zhuyuanqing@hrbeu.edu.cn

**Abstract:** The gradual opening of the Arctic shipping route has made navigation possible. However, the harm caused by ship exhaust emissions is increasingly severe. Therefore, it is necessary to study the diffusion characteristics of ship exhaust plumes during Arctic navigation. The study focuses on a merchant vessel as the subject of investigation, employing computational fluid dynamics (CFD) simulation techniques to analyze the diffusion characteristics of particulate matter (PM) within ship exhaust plumes under Arctic environmental conditions. The diffusion law of ship exhaust plume PM is clarified, and the influence of three factors, synthetic wind speed, yaw angle and chimney angle, on the PM diffusion is analyzed. It was found that after the PM was discharged from the chimney, the majority of the PM dispersed directly backward along with the external flow field, while a minor fraction lingered at the stern of the ship for an extended period before eventually diffusing backward. Among them, 1235 particles were captured within a range of 200 m from the stern, with a capture rate of 0.6%. When the synthetic wind shows a yaw angle, the capture rate of PM in the interval increases rapidly with the increase of yaw angle, while other factors have less influence on the capture rate of PM. This study provides foundational guidance for predicting PM diffusion from ship exhaust plumes in Arctic environments, thereby enabling more effective strategies for managing such emissions.

**Keywords:** arctic sea lanes; diffusion characteristics; particulate matter; ship exhaust

## 1. Introduction

In terms of climate change, the Arctic is a highly uncertain environment on Earth [1]. With global warming, the sea ice coverage in the Arctic has been decreasing year by year since the end of the 20th century, especially in the summer and autumn seasons, so that commercial navigation in the Arctic shipping routes has become possible [2,3]. The Arctic shipping route is of great strategic value. As a sea shortcut connecting the Pacific and Atlantic Oceans, the Arctic shipping route will dramatically shorten the distance of maritime transportation compared to traditional shipping routes. In recent decades, the Arctic shipping window for all types of ships has increased significantly, and it is expected that Polar Class 6 (PC6) ships will be navigable year-round around the 2070s when the decadal-averaged global mean surface temperature anomaly hits approximately +3.6 °C compared to pre-industrial times (1850–1900) [4–6].

Although the Arctic is located far away from midlatitude pollution sources, air pollution is pervasive in the Arctic [7]. With the increase in the number of ships sailing in the Arctic, the air pollution caused by ship exhaust is becoming more and more serious. In particular, PM gained attention recently and was the subject of various studies about ship emissions [8–10]. Particle matter (PM) refers to solid or liquid PM dispersed and floating in the atmosphere, containing complex components [11]. Particles emitted from ships impact climate through

both direct and indirect effects and are often emitted close to populated coastlines where they impact air quality [12,13]. PM affects the climate indirectly by modifying the optical and microphysical properties of clouds and acting as cloud condensation nuclei (CCN). As a consequence, long and narrow clouds called ship tracks are formed behind a ship so that heavily frequented ship routes can even be observed on satellite images [14,15]. As the global atmosphere flows, PM adheres to the surface of glaciers in polar regions, reducing the reflection of sunlight from the glaciers, raising the surface temperature of the glaciers, and thus accelerating the melting of the glaciers. The PM emitted by ship engines consists of organic carbon (OC), elemental or black carbon (EC/BC), sulfate, inorganic compounds containing V, Ni, Ca, Zn and other metals and sulfate-associated water [16,17]. Black carbon is a unique carbonaceous substance that only forms in flames during the combustion of carbon-based fuels [18]. In the Arctic, especially in summer, black carbon has significant light absorption and low surface albedo. BC impacts the Arctic climate both in the atmosphere and upon being deposited onto ice and snow. The greenhouse effect of black carbon, measured in terms of the effect of raising surface air temperature per unit forced, is twice that of carbon dioxide, and it may be more effective at melting ice and snow [19]. The shipping industry has been developing towards zero carbon emission rapidly [20]. Therefore, in order to continue promoting the development of green shipping in the Arctic, it is necessary to study the diffusion laws of PM.

The emission characteristics of PM from ships depend on various parameters, such as particle number, particle mass concentration, particle size and particle chemical composition [21]. Most particles emitted from ships are in the sub-micrometer range, typically with a diameter below 100 nm [22]. Cappa et al. [23] found that the measured EFs (fuel mass basis) for PM1 mass, BC and POM decreased as the ship slowed, and the particle number EFs were approximately constant across the speed change. Sinha et al. [24] measured the particulate and gas emissions from two diesel-powered ships and discussed the effects of fuel grade and engine power on ship emissions.

At present, there has been some research on the emission components of PM from marine diesel engines. Shen and Li [25] found that the sulfate emissions from marine diesel can be ignored, while marine diesel is purely dominated by organic carbon and elemental carbon. Zetterdahl et al. [26] studied the influence of different fuels on PM of two-stroke diesel engines on ships and concluded that low-sulfur fuels could reduce the emission quality of PM, but there was no significant change in EC emission. Ausmeel et al. [27] used a new method based on wind field data and ship Automatic Identification System (AIS) data to measure atmospheric particles and realize the identification of individual ship plumes.

In past years, several studies on particle emissions from shipping and their effects on marine stratus clouds were conducted. Approaches include test rig measurements or exhaust stack sampling, airborne and shipborne measurements in ship plumes, combinations of exhaust studies and plume studies using airborne platforms and finally, the investigation of the CCN fraction of particles, which potentially form cloud droplets [28]. To date, a number of atmospheric studies of individual ship plumes have been conducted. Among them, CFD provides the research method of visualization of exhaust emission path optimization. Kulkarni et al. [29] studied the interaction between smoke exhaust and ship superstructure. The effectiveness of CFD simulation of ship chimney exhaust plumes was demonstrated by comparing the CFD calculation results with the measured results. Xu et al. [30] analyzed the exhaust plume characteristics of container ships with different stack numbers using CFD software, and the results showed that the transmission and diffusion characteristics of the multi-chimney exhaust were different from those of single-chimney exhaust. Kim et al. [31] conducted a CFD flow field simulation and a comparative study of ship exhaust diffusion on cargo ships to optimize the shape design of cargo ship chimneys. Zhou et al. [32] developed a UAV-based ship exhaust measurement system and detection process to achieve accurate tracking and measurement of ship plumes. Lee et al. [33] confirmed the emission trend of pollutants in the exhaust gas of

the engine under different working conditions by using the exhaust gas analyzer. The results showed that as the sampling position moved from the turbocharger to the chimney, more condensate was observed at low temperatures, and the molecular structure of PM gradually appeared as an amorphous structure. Murphy et al. [34] reported the first study to simultaneously install and measure emissions from a container ship, evaluating the relationship between shipborne and airborne measurements and exploring how these findings affect the emission characteristics of such ships.

Currently, research on the diffusion of PM emissions from ship plumes mainly focuses on conventional sea areas, while there is relatively little research in Arctic ice regions. Ships operating in the Arctic are likely running at highly variable engine loads (25–100%) depending on ice conditions and ice-breaking requirements. The ships operating at low loads may be emitting up to 50% more BC than they would at their rated load. The variability in load conditions complicates the assessment of potential BC emission rates [35]. The cold and statically stable marine boundary layer in the Arctic, which is governed by the effects of surrounding ice and small changes in solar zenith angle, is likely to impact the dispersion and expansion of the ship plumes differently [36]. Therefore, ship plume prediction models for the Arctic region, compared with those for mid-latitude areas, exhibit distinct boundary layer dynamics.

Considering the impact of PM on health and the environment in the Arctic region, this study takes a merchant ship as the research object and uses CFD simulation as the calculation method to conduct a study on the diffusion of PM in the Arctic region. The diffusion law of PM in ship exhaust under low-temperature environments is clarified, and the influence of different factors on the diffusion of PM in the plume is studied. This study provides guidance for further reducing the impact of PM on the atmosphere in the Arctic region.

## 2. Numerical Calculation Method

### 2.1. CFD Mathematical Model Establishment

The three fundamental constraints in fluid flow—mass, energy and momentum conservation—form the core control equations for fluid dynamics. The thermodynamic state of ship exhaust diffusion involves both the gas and solid phases, for which the standard $k$-$\varepsilon$ model, the component transport model and the discrete-phase model (DPM) were chosen to simulate the diffusion of exhaust. Among them, the standard $k$-$\varepsilon$ model is used to simulate the flow diffusion of particles and the flow of the external air flow field; the component transport model is used to describe the gaseous composition of the exhaust, and the discrete-phase model is used to simulate the diffusion of the carbon soot particles [37–39].

In the component transport model, Fluent predicts the mass fraction $Y_i$ of each component by the convection–diffusion equation for the ith substance. The general form of the conservation equation is as follows.

$$\frac{\partial}{\partial t}(\rho Y_i) + \nabla \cdot (\rho v Y_i) = -\nabla J_i + R_i + S_i \tag{1}$$

In the formula, $R_i$ represents the net rate of chemical reaction produced by the ith component; $S_i$ represents the additional rate caused by discrete phase and user-defined source terms; $Y_i$ represents the mass fraction of the ith component.

In the discrete-phase model, Fluent software establishes control equations in the Lagrangian coordinate system to calculate and simulate the forces acting on the discrete phase, analyze the motion laws of particles and track their trajectory. The descriptive equation is as follows.

$$\frac{du_p}{dt} = F_D + \frac{g(\rho_p - \rho)}{\rho_p} + F_x \tag{2}$$

$$F_D = \frac{u - u_p}{\tau_r}, \tau_r = \frac{\rho_p d_p^2}{18\mu} \cdot \frac{24}{C_d Re} \tag{3}$$

$$Re \equiv \frac{\rho d_p |u_p - u|}{\mu}, C_d = a_1 + \frac{a_2}{Re} + \frac{a_3}{Re^2} \tag{4}$$

where $F_D$ represents the drag force acting on the PM [N]; $F$ represents other external forces [N]; $\tau_r$ represents the relaxation time of the PM [s]; $Re$ represents the relative Reynolds number; $C_d$ represents a constant, $u$ represents the velocity of the exhaust gas [m/s]; $u_p$ represents the velocity of the PM [m/s]; $\mu$ represents the viscosity coefficient of the exhaust gas [Pa·s]; $\rho$ represents the density of the exhaust gas [kg/m$^3$]; $\rho_p$ represents the density of the PM [kg/m$^3$]; $d_p$ represents the diameter of the PM [m].

### 2.2. Parameter Definition of the Diffusion Effect of Ship Exhaust Particulate Matter

#### 2.2.1. Yaw Angle and Synthetic Wind Speed

During the navigation of a ship, the natural wind direction is often different from the ship's navigation direction, so the relative angle between the ship's heading and the natural wind direction is defined as the yaw angle ($\theta_1$), as shown in Figure 1.

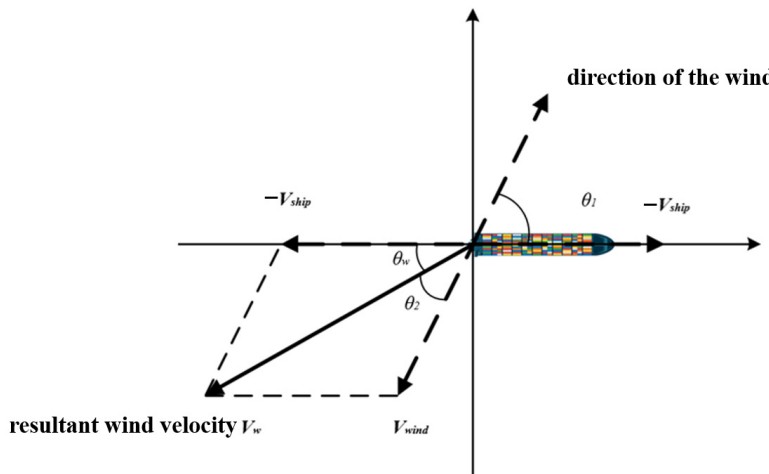

**Figure 1.** Wind speed composite vector diagram.

In the figure, the ship's movement direction is taken as the positive *X*-axis direction. Due to the fact that the direction of wind flow is opposite to the direction of wind, and the relative direction of airflow caused by ship movement is also opposite to the ship's heading. If the ship is used as the reference frame, then the synthetic wind speed ($V_w$) is the vector sum of the opposite value of the ship's sailing speed ($-V_{ship}$) and the natural wind speed ($V_{wind}$). At this point, the yaw angle of the synthetic wind speed is $\theta_w$. The direction of the synthetic wind speed is the theoretical direction of exhaust gas diffusion.

#### 2.2.2. Chimney Angle

The chimney of a ship includes the main engine chimney and the auxiliary engine chimney. In this study, only exhaust emissions from the main engine stack are considered. Through research, it has been found that the angle of the chimney varies in different ship designs. As shown in Figure 2, the stern direction is the positive direction of the *X*-axis, and the vertical upward direction is the positive direction of the *Y*-axis. The inclination angles of the chimney are all in the first quadrant, so this study stipulates that the angle of the chimney is the degree of the angle between the chimney's central axis and the *X*-axis, and the counterclockwise direction is the positive direction without considering the negative angle.

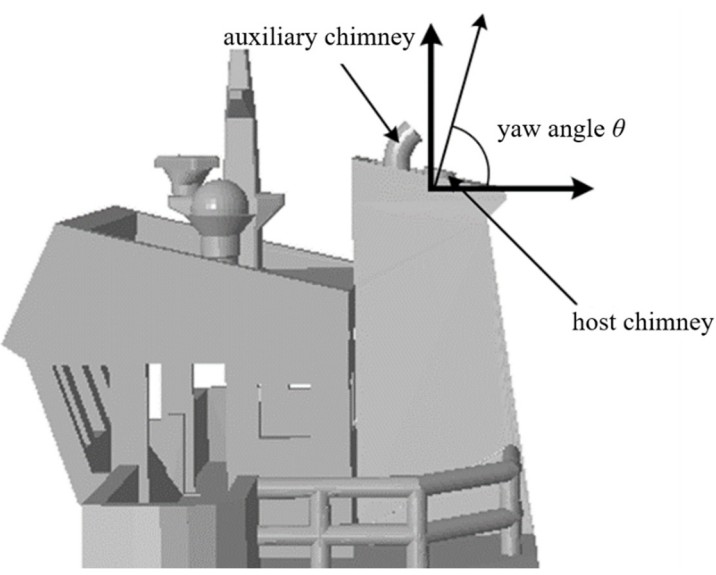

**Figure 2.** Chimney angle definition diagram.

*2.3. Computational Model and Mesh Division*

In this study, a merchant ship with a load capacity of 15,000 tons is selected as the research object. Its specific dimensions are shown in Table 1.

**Table 1.** Ship size parameters.

| Tonnage of Ship DWT (t) | Design Ship Size (m) | | | |
|---|---|---|---|---|
| | **Length** | **Breadth** | **Depth** | **Load Draught** |
| 15,000 (12,501~17,500) | 153 | 23 | 12.9 | 9.4 |

According to the selected ship type and ship size, a three-dimensional model of the merchant ship is established, as shown in Figure 3. However, considering comprehensive factors such as simulation calculation quantity and mesh quality, the detailed features of the ship are simplified and only the part of the ship exposed to the air is considered. The simplified model is shown in Figure 4.

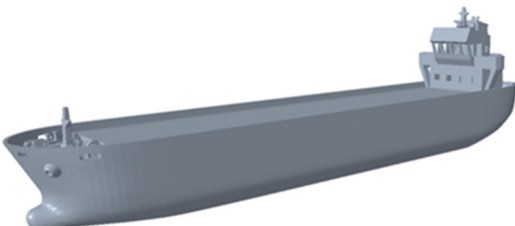

**Figure 3.** Merchant ship 3D model.

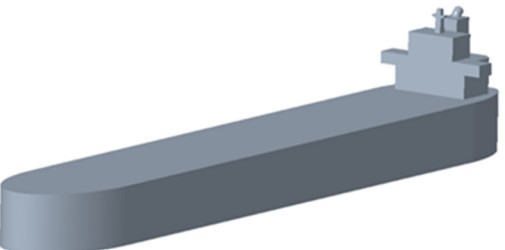

**Figure 4.** Simplified merchant ship 3D model.

In this study, the external flow field calculation domain adopts a length of 600 m (4 times the length of the ship), a width of 400 m (20 times the width of the ship) and a height of 150 m (10 times the height of the ship). The fluid computational domain is divided into near-field computational domain and far-field computational domain. A rectangular area of 20 m around the ship is selected as the near-field watershed, and the rest of the area is the far-field area. The computational domain and the ship model use Boolean operations to remove the solid part of the ship, leaving only the air computational domain and treating the ship wall and water surface as rigid surfaces. The divided computational domain is shown in Figure 5.

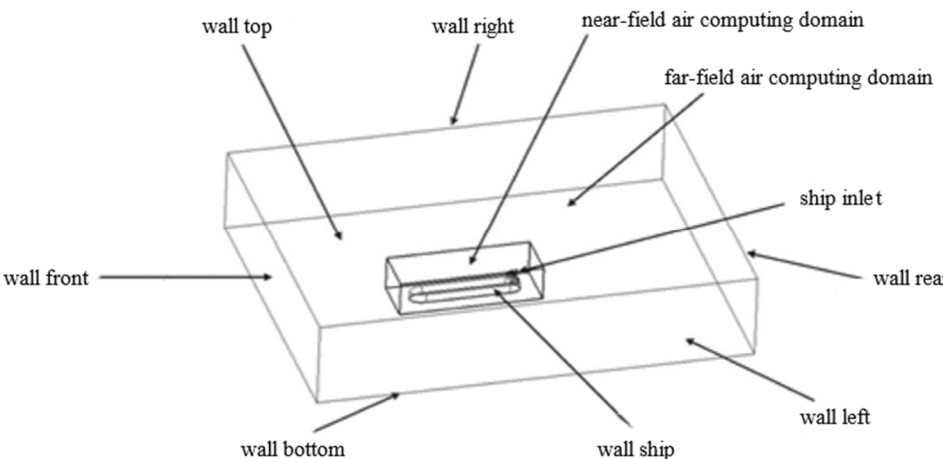

**Figure 5.** Computational domain partitioning diagram.

In determining the model, although the hull shape has been partially simplified, there are still certain areas that are excessively large or relatively small in size. Therefore, it is necessary to adopt a segmented mesh division mode for the overall calculation domain, as shown in Figure 6.

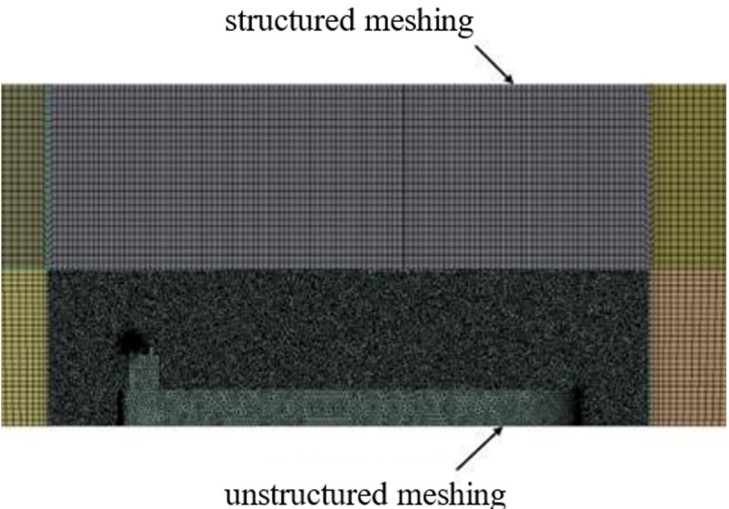

**Figure 6.** Computational domain mesh partitioning.

The mesh used in this study is a joint partitioning method of structured and unstructured mesh. The structural and unstructured mesh are connected by a common node to ensure effective data transmission between mesh. The structural cell control size is 3 m, and the unstructured cell control size is 1 m. The number of cells is in the range of 7–9 million (due to differences in chimney angles). The average quality of the mesh is

0.877, and the average value of the mesh torsion is 0.132, which meets the requirements of the calculation.

### 2.4. Mesh Independence Verification

In this study, the determination of mesh independence is mainly based on the fluctuation of import and export flow and the difference in import and export flow. Due to the fact that this study mainly studies the computational domain of exhaust gas near ships, and the boundaries of the computational domain near ships are relatively complex, which requires high mesh requirements, this section mainly focuses on verifying the mesh independence of the near-field air computational domain. The far-field mesh adopts a triple magnification size to meet the computational requirements.

In terms of the selection of cell size, this section verifies that the cell size is 2 m, 1.5 m, 1 m and 0.7 m, which decreases successively. The calculation requires the setup of exhaust inlet mass flow monitoring, air inlet mass flow monitoring and outlet mass flow monitoring. The converged data were analyzed to verify the fluctuation of import and export flow and the flow difference. The trend of the above two parameters with the refinement of cell size is shown in Figure 7.

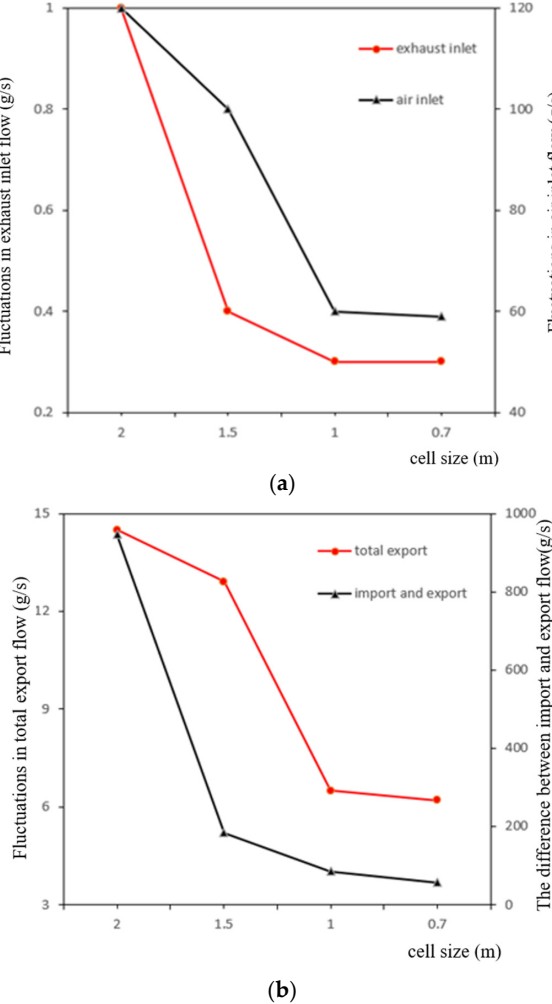

**Figure 7.** Fluctuation chart of flow rate at each boundary. (**a**) Fluctuation chart of exhaust and air inlet mass flow rate. (**b**) Fluctuation chart of total export and import/export mass flow rate.

As can be seen in Figure 7, the flow fluctuations at each boundary port and the difference in flow at the import and export decrease with smaller cell sizes. After the cell size reaches 1 m, the fluctuation values of the mass flow at the exhaust and air inlet

basically remain unchanged. After the cell size reaches 1 m, the fluctuation value of the total export mass flow rate and the difference in import and export mass flow rate also decrease gradually. The total number of cells for the 2 m cell size is 3 million, the total number of cells for the 1.5 m cell size is 7 million, the total number of cells for the 1 m cell size is 12 million and the number of cells for the 0.7 m cell size is at 16 million. After the number of cells reached 10 million, the workstation computation hours increased dramatically. Therefore, a cell size of 1 m is selected as the final calculation cell size, taking into account multiple factors such as computational accuracy and workload.

*2.5. Boundary Conditions and Parameters Determination*

In practice, the navigation of ships at sea is complex, and the shape of ships is influenced by various factors such as waves and strong winds. Considering that the environment applicable to this model is the pollution discharge domain of ships near the Arctic region, the boundary conditions selected for the simulation model in this study refer to the conditions of good environmental weather and smooth operation of ships in the Arctic region near the sea. In the calculation process, the influence of the water surface on the air flow field is ignored, and the liquid surface is set as a flat and rigid surface. The parameters, such as draft and deflection angle, of the ship remain unchanged.

In the calculation process, the following ideal assumptions are made for some of the calculation conditions: (1) it is assumed that the wind speed remains uniform and has no relationship with the altitude above the ground; (2) it is assumed that the ship speed is uniform and the main engine exhaust speed is uniform; (3) it is assumed that the air pressure at sea is one atmosphere (101,325 Pa), the temperature is 20 °C (293 K) and the gas component is normal air, which is not affected by seawater evaporation; (4) it is assumed that particles are not affected by adsorption on the ice surface.

In this simulation, since air temperature and exhaust temperature have little effect on the diffusion of PM [40], the effect of its change will not be considered. In this study, the atmospheric temperature in the Arctic shipping lanes is set to −5 °C [41], and the exhaust gas temperature is set to 100 °C for normal navigation. Due to the high wind speed in the Arctic, 10 m/s is used as the base wind speed for research in this simulation. Because ships are distributed symmetrically in their heading direction, only unilateral yaw angles are calculated when considering yaw angles. Because the airflow brought by the ship's movement always flows towards the stern, the range of yaw angle for the defined synthetic wind is 0° to 90°. Due to the fact that the inclination angle of the chimney is in the first quadrant, the range of chimney angle variation is from 0° to 90°.

The emission and proportion of PM mainly refer to the emission situation of heavy oil combustion. The Rosin Rammler distribution method is used to interpolate the size distribution to satisfy the characteristics of different emission diameters without considering the growth of particles. In academia, particles are usually classified into nuclear modes (diameter 3–20 nm), Aitken nuclear modes (diameter 20–90 nm), accumulation modes (diameter 90–1000 nm) and coarse particle modes (diameter greater than 1000 nm) based on their diameter size [42]. Moldanová et al. [43] analyzed the distribution characteristics of PM emissions from large ships and found that when using heavy oil, most of the PM emitted from internal combustion engines is nuclear-mode particles in diameter. In the simulation process, in order to simulate the effect of particulate deposition, the computational and bottom boundaries are set as the particulate capture option, the hull is set as particulate rebound, and all other boundaries are set as particulate escape. The specific condition boundaries are shown in Table 2.

**Table 2.** Boundary conditions in the Arctic environment.

| Air Velocity | Air Temperature | Exhaust Velocity | Exhaust Temperature | Particle Velocity | Particle Mass Flow | Particle Ejection Temperature |
|---|---|---|---|---|---|---|
| 10 m/s | 268 K | 10 m/s | 373 K | 10 m/s | 0.05 kg/s | 373 K |

### 2.6. CFD Mathematical Model Verification

This study adopts the paper comparison verification method for verification. By selecting a public paper containing experimental verification that is similar to the simulation content in this study, the simulation calculation is carried out according to the experimental and simulation boundary conditions and the calculation model to be verified. After obtaining the calculation results, we compare them with the simulation results and experimental results of the paper to confirm the usability and accuracy of the model. In this study, two papers, including simulation and experiment contents, are selected as the comparison verification group, and two cases are selected for each group of papers as the control.

The first paper selected for this verification is titled "*Numerical Modeling of Exhaust Smoke Dispersion for a Generic Frigate and Comparisons with Experiments*" by Ergin et al. [44] from Istanbul Technical University. The paper focuses on exhaust diffusion from a general-purpose frigate. The speed ratio (exhaust gas speed/synthetic wind speed) K in this study is about 0.6, so two cases with a speed ratio K of 0.407 and 0.815 in the paper are selected for comparison, which is not significantly different from the speed ratio in this study. Their simulation comparison is shown in Figures 8 and 9.

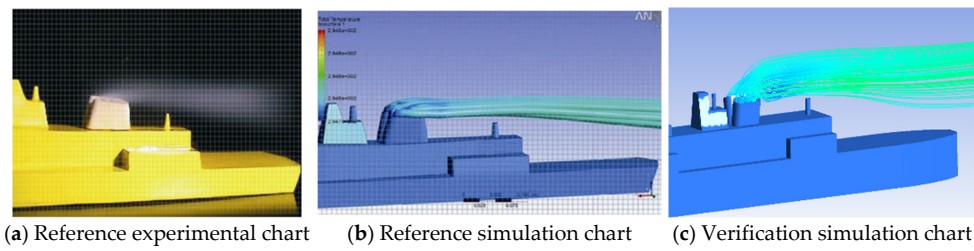

(**a**) Reference experimental chart    (**b**) Reference simulation chart    (**c**) Verification simulation chart

**Figure 8.** The speed ratio K is 0.407 for verification comparison [44].

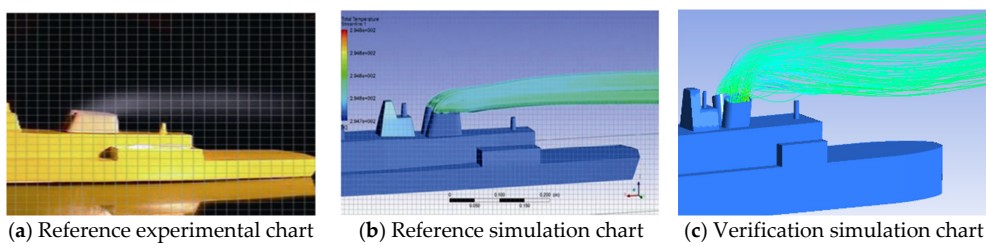

(**a**) Reference experimental chart    (**b**) Reference simulation chart    (**c**) Verification simulation chart

**Figure 9.** The speed ratio K is 0.815 for verification comparison [44].

Since the actual size and simulated boundary conditions of the frigate model are not given in this paper, reverse modeling is carried out according to the conventional size of the frigate and the ratio method. The case's similar boundary conditions are given according to the exhaust gas characteristics and speed of the frigate. As can be seen from Figure 8, when the velocity ratio is low, the experimental results, simulation results of the paper and simulation results of this study all show exhaust gas sinking, and the angle of exhaust gas sinking in this study is more consistent with the experimental conclusion. When the speed is relatively high, as shown in Figure 9, the exhaust gas is parallel and backward diffused in both the experimental conclusion and the simulation conclusion of the paper. In this study, the simulation results of the exhaust gas first show a tendency of partial sinking but then diffuses in parallel backward, which is consistent with the experimental results of the paper. The main reason for the error comes from the slight difference between the size of the simulation model used in this study and the size used in the paper. In the comparative experiment conducted in this study, it is found that the height of the frigate tower is the main factor affecting the trajectory of plume diffusion. Therefore, this study concludes that this controlled experiment is valid.

The second paper selected for this verification is titled "*Parametric studies of exhaust smoke-superstructure interaction on a naval ship using CFD*" by P.R. Kulkarni [45] from the Indian Institute of Technology. The paper mainly studied the exhaust diffusion of four

types of ship superstructure under different speed ratios and wind directions. This study selects the two cases in the paper that are closest to the speed ratio K in this study, namely the two cases with K being 1 and 2, for simulation comparison.

The author of this paper provides detailed simulation model size parameters, as shown in Figure 10, which enables better simulation reproduction. The case selected in this study is a simplified model of a ship with dual exhaust gas, in which a mast is in front of the exhaust pipe of a ship on the front side, which affects the exhaust gas diffusion. The results are shown in Figures 11 and 12. As can be seen from Figure 11a, when the velocity ratio K is equal to 1, the low-pressure area generated behind the mast causes the exhaust gas to wash the mast after discharging from the front chimney and then impact and spread. The exhaust gas diffuses over a large area at a height higher than the horizontal line of the mast, flows backward, converges with the rear exhaust gas and diffuses backward together. In the simulation results of the paper, although the exhaust gas from the front chimney appeared to wash towards the mast, it can be seen from the streamline in the figure that only a small portion of the exhaust gas was at the edge, and most of the exhaust gas still diffused towards the rear. The exhaust did not diffuse to the area higher than the mast, resulting in significant differences in simulation results. By observing the experimental phenomenon, it can be seen that the simulation results in this study have a higher degree of agreement, and their streamlined trajectory is consistent with the diffusion law of exhaust gas in the experiment. Therefore, the credibility of the results in this study is relatively high.

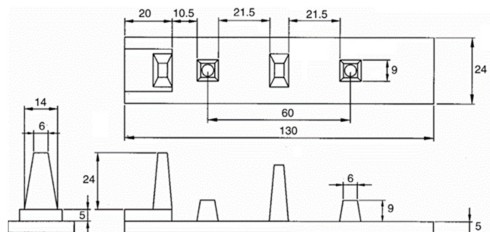

**Figure 10.** Dimensions of each part of the simulation model (in centimeters) [45].

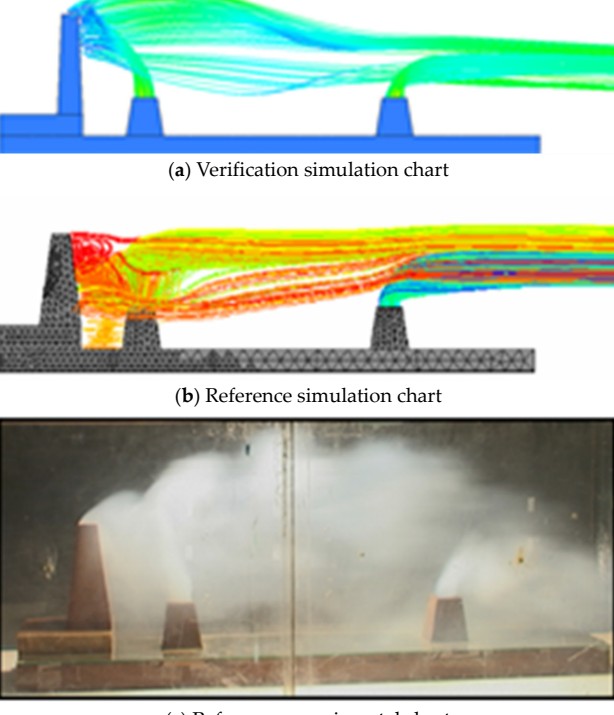

(**a**) Verification simulation chart

(**b**) Reference simulation chart

(**c**) Reference experimental chart

**Figure 11.** The speed ratio K is 1 for verification comparison [45].

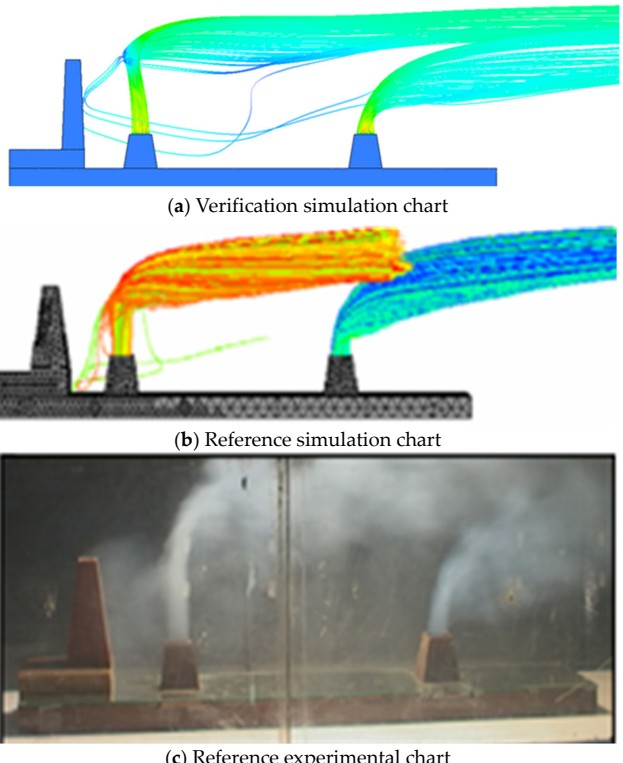

(**a**) Verification simulation chart

(**b**) Reference simulation chart

(**c**) Reference experimental chart

**Figure 12.** The speed ratio K is 2 for verification comparison [45].

When the speed ratio K is equal to 2, the simulation results in this study, the simulation results in the paper, and the experimental results are relatively consistent, as shown in Figure 12. Therefore, this study believes that the controlled experiment is valid.

Through the comparison and simulation of the above two papers, it can be concluded that the computational model used in this study has high accuracy and can meet the credibility of simulation calculations.

## 3. Simulation Results and Analysis

### 3.1. Diffusion Law of Particulate Matter in Ship Exhausts

According to the particle tracking diagram in Figure 13, after the particles are discharged from the chimney, most of the particles directly diffuse to the rear along with the external flow field, while a small number of particles still remain at the stern of the ship for a long time and then diffuse to the rear. According to the analysis, the reason for this phenomenon is that the air will form eddy currents at the back side of the ship when it flows through the outer surface of the ship. Some of the particles are caught in the vortex during the diffusion process, resulting in a longer period of stagnation. The particles will rotate with the vortex for a long time.

By observing the longitudinal particle distribution diagram, it can be seen that some particles will hit the water face behind the ship during the diffusion process. With particle tracking, 192,327 particles were tracked. At a range of 200 m astern, 1235 particles were captured, accounting for 0.6% of the tracked particles. These captured particles will adhere directly to the ice or snow, affecting the melting of glaciers.

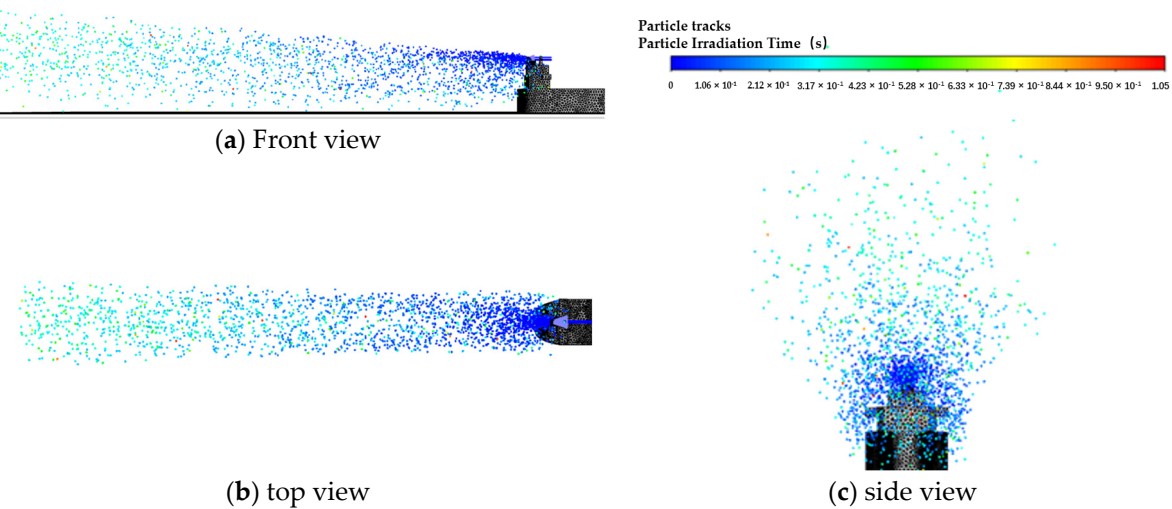

**Figure 13.** Diffusion time distribution of particulate matter (PM).

*3.2. Analysis of the Influence of Synthetic Wind Speed on the Diffusion of Particulate Matter from Ship Exhaust*

In this section, four cases with synthetic wind speed ranging from 10 m/s to 17.5 m/s are calculated. The wind speed increases equally in sequence while other boundary conditions remain unchanged. The specific parameters are shown in Table 3.

**Table 3.** The corresponding synthetic speed of each example.

| Case Name | Case 1 | Case 2 | Case 3 | Case 4 |
|---|---|---|---|---|
| Synthetic wind speed | 10 m/s | 12.5 m/s | 15 m/s | 17.5 m/s |

It can be seen from Figure 14 that with the increase in velocity, the overall diffusion mode and range of PM do not change greatly. But at the stern of the ship, it can be seen that as the wind speed increases, the range of particle diffusion partially increases. The reason for this phenomenon is that when the wind speed increases, the vortex area generated by the stern is larger in volume and has more turbulent kinetic energy. Therefore, when particles are sucked into the vortex, their range of motion is larger.

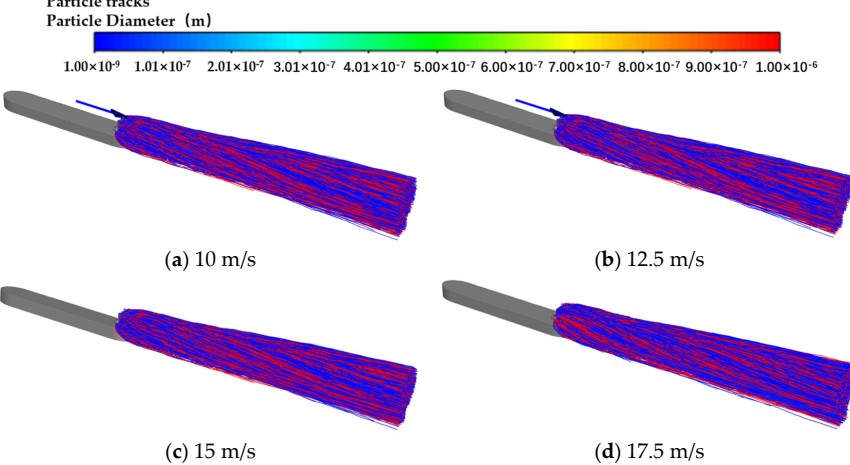

**Figure 14.** Particle diffusion trajectories at each synthetic wind speed.

As can be seen from Figure 15, the particle capture rate increases with the increase in synthetic speed. Among them, the speed increases the fastest in the range of 10 m/s to

12.5 m/s, followed by a slow decline. However, the overall particle capture rate (number of captured particles/number of tracked particles) changes little. Through analysis, it can be seen that within the range of 10 m/s to 12.5 m/s, it is near the critical value of the external flow field for exhaust gas diffusion. Therefore, as the velocity increases, the chaos degree of the exhaust diffusion increases considerably so that the particle capture rate rises rapidly in this interval and then slowly decreases. Therefore, it can be concluded that when the stability of the external flow field is worse, the number of particles captured by the ice surface is higher. In order to reduce the impact of PM on Arctic ice melting, ships should try to navigate in an environment with lower synthetic wind speed.

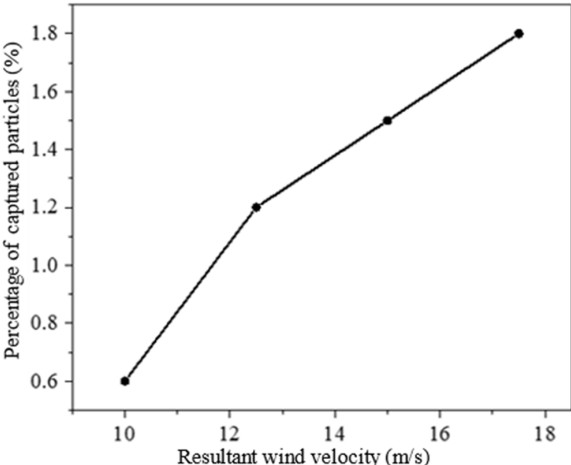

**Figure 15.** Particle capture rate at each synthetic wind speed.

*3.3. Analysis of the Influence of Yaw Angle on the Diffusion of Particulate from Ship Exhaust*

In this section, four cases with yaw angles ranging from 0° to 90° are calculated. The deflection angle increases equally in sequence, while other boundary conditions remain unchanged. The specific parameters are shown in Table 4.

**Table 4.** The corresponding yaw angles of each example.

| Case Name | Case 1 | Case 2 | Case 3 | Case 4 |
|---|---|---|---|---|
| Yaw angle | 0° | 30° | 60° | 90° |

It can be seen from Figure 16 that with the increase in the angle, the diffusion trajectory of PM is gradually confused. When deflection occurs, the main diffusion region of PM shows a downward trend. When the yaw angle reaches 90°, due to the large blocking effect on the incoming flow caused by both the hull and the ship building, some particles diffuse along the airflow to the side of the hull, and some particles diffuse above the deck. The density of the streamlines shows that only a portion of the PM diffuses with the airflow to the bow and deck, while most of the PM diffuses towards the direction of the airflow as soon as it excludes the chimney.

It can be seen from Figure 17 that the capture rate of PM on the ice surface increases significantly when the direction of the incoming flow is deflected. When the yaw angle reaches 60°, 19 percent of all particles are captured. However, as the angle continues to increase, the capture rate of PM decreases slightly.

When the projected area is larger and more irregular, the vortex appearing on the leeward side of the ship is more chaotic. If the chimney outlet is located in an area with high turbulent energy, the diffusion of exhaust gas is more chaotic, and the probability of particles colliding with the ice surface is higher, resulting in a higher number of captured particles. When the yaw angle is 60°, the chimney outlet is still within the projected area of the ship building. Compared to a 30° yaw angle, the projection area is larger, so the particle

capture rate rapidly increases. When the yaw angle is 90°, although the projected area is the largest at this time, the chimney outlet is also right at the edge of the projection. At this time, the direction of the exhaust velocity is far away from the projection area so that the influence of the vortex on the exhaust at the leeward side of the hull is reduced. Only a small portion of the PM diffuses along with the airflow to the bow and deck, while most of the PM diffuses directly into the air.

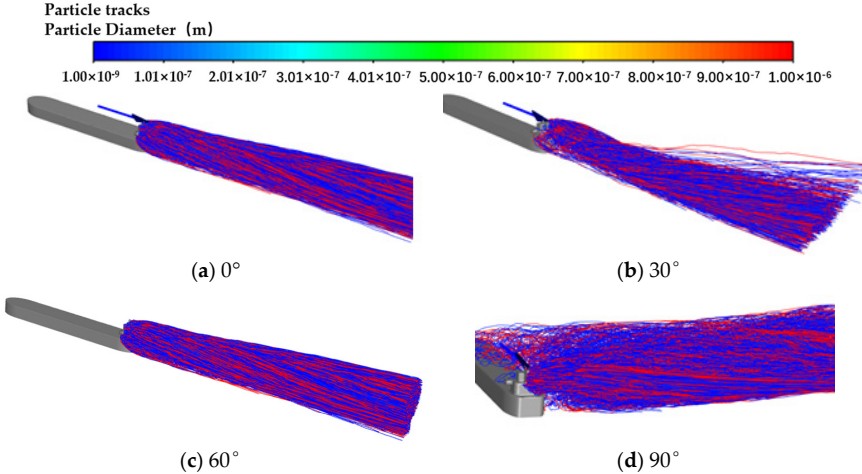

**Figure 16.** Particle diffusion trajectories at each yaw angle.

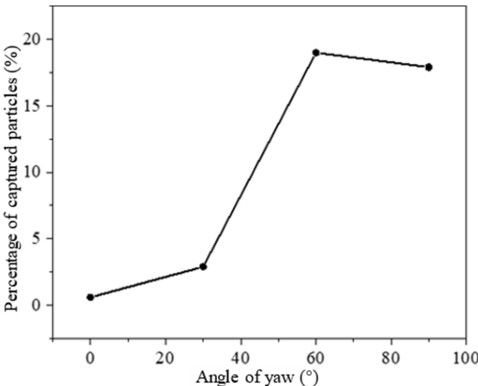

**Figure 17.** Particle capture rate at each yaw angle.

Since the capture rate of PM is the highest when the yaw angle is 60°, in order to reduce the melting effect of PM on the Arctic ice, the yaw angle of the ship should be deviated from 60° as far as possible during navigation.

*3.4. Analysis of the Influence of Chimney Angle on the Diffusion of Particulate from Ship Exhaust*

In this section, four cases with chimney angles ranging from 0° to 90° are calculated. The deflection angle increases equally in sequence, while other boundary conditions remain unchanged. The specific parameters are shown in Table 5.

**Table 5.** The corresponding chimney angles of each example.

| Case Name | Case 1 | Case 2 | Case 3 | Case 4 |
| --- | --- | --- | --- | --- |
| Chimney angle | 0° | 30° | 60° | 90° |

According to Figure 18, the chimney angle does not show significant changes in the overall trajectory of PM diffusion, but there are differences in the area near the stern. By observing the trajectory diagram of the stern part, it can be seen that as the chimney

angle rises, the diffusion area of the exhaust gas also slowly moves away from the ice. After contact with the ice surface, the exhaust gas diffuses backward in a ship's projected trajectory with no significant difference.

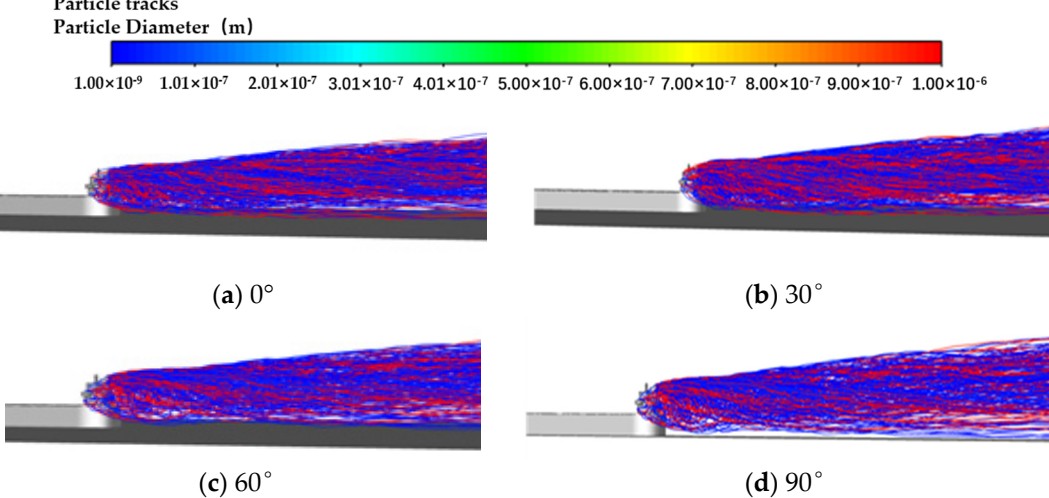

**Figure 18.** Particle diffusion trajectories at each chimney angle.

As can be seen from Figure 19, the particle capture rate gradually decreases with the increase of the chimney angle. At the same angle of improvement, the reduction value of the capture rate of the particle gets smaller and smaller. As the angle of the chimney rises, the initial velocity angle of the exhaust also shows an elevation angle. However, due to the influence of the external flow field, the energy of the exhaust is quickly depleted, so only a small difference appears in the stern of the ship. As the chimney angle increases, the diffusion area of the exhaust gases in the stern section increases relatively, which results in a relatively smaller particle contact area in the same ship side area and, thus, a lower particle capture rate. When the chimney angle rises from 0° to 90°, the capture rate decreases by only 0.06%. Therefore, the chimney angle has no significant effect on the diffusion of particles. In order to reduce the amount of particles attached to the ice surface, it is recommended that the chimney angle of the ship be close to 90° during navigation in the Arctic.

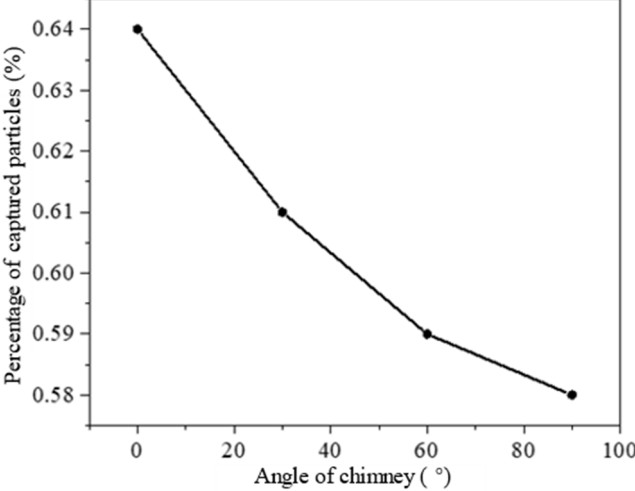

**Figure 19.** Particle capture rate at each chimney angle.

## 4. Conclusions

With the development of the shipping industry in the Arctic environment, the environmental pollution caused by PM from ship exhaust has become more and more serious. Countries around the world have established emission control zones to limit the emission of PM from ship exhaust. However, there is currently no good method for predicting the diffusion trajectory of PM in ship exhaust plumes. In response to the above issues, this study takes a merchant ship as the research object and uses CFD simulation as the calculation method to conduct a study on the diffusion of PM in ship exhaust plumes in the Arctic environment. The diffusion law of PM in ship exhaust in the Arctic environment is studied, and the influence of various factors on the diffusion of PM in ship exhaust is explored. In addition, based on the diffusion law, a more suitable synthetic wind speed, yaw angle and chimney angle for the merchant ship in this study are proposed to reduce the impact of PM on melting Arctic ice.

The results show that most of the particles are directly diffused to the rear with the external flow field after being discharged from the chimney, but a few particles still stay in the stern for a long time and then diffuse to the rear. When the synthetic wind speed is higher, the percentage of PM captured by the ice surface within the same range is higher, but the overall impact is smaller. When the yaw angle appears in the synthetic wind, the particle capture rate increases rapidly with the increase of the yaw angle in the range of 0° to 60°, but the particle capture rate decreases slightly in the range of 60° to 90°. The yaw angle has a great influence on the diffusion of PM. The capture rate of PM slowly decreases with the increase of chimney angle, but the overall impact of chimney angle on the diffusion and capture of PM is relatively small.

Among the three factors studied, yaw angle has the greatest influence on the diffusion of PM. Therefore, in the actual Arctic navigation, in order to effectively reduce the environmental pollution caused by the emission of PM, it is a good idea to control the yaw angle. Although some laws have been summarized in this study, no theoretical research results have been formed. Subsequent work may continue to further study solutions to reduce PM emissions.

This study selects two public papers that are essentially similar to the simulation content of this study and verifies the numerical models used in the research. The results show that the simulation results are in good agreement with the experimental results. This study indicates that computational fluid dynamics (CFD) is a powerful tool for studying the diffusion problem of ship exhaust plumes. Furthermore, this study provides a way to further understand the diffusion of PM in ship exhaust plumes in the Arctic environment.

**Author Contributions:** Conceptualization, Y.Z., Y.F. and C.X.; Methodology, Y.Z., Q.H., Y.F., J.S. and C.X.; Software, Y.Z., Q.W., Q.H. and J.S.; Validation, Y.Z., Q.W. and Q.H.; Formal analysis, Y.Z., Q.W., Q.H., Y.F., J.Y. and C.X.; Investigation, Y.Z., Q.W., Q.H. and J.Y.; Resources, Y.Z., Q.W., Q.H., J.Y. and C.X.; Data curation, Y.Z., Q.W., Q.H., J.Y. and J.S.; Writing—original draft, Q.W. and Q.H.; Writing—review & editing, Y.Z. and Y.F.; Visualization, Y.Z. and Y.F.; Supervision, Y.Z. and Y.F.; Project administration, Y.Z.; Funding acquisition, Y.Z. All authors have read and agreed to the published version of the manuscript.

**Funding:** The authors disclosed receipt of the following financial support for the research, authorship, and/or publication of this article. The work was supported by the Fundamental Research Funds for the Central Universities of China [grant number 3072022JC0305] and the Excellent Youth Science Foundation of Hei Longjiang Province [grant number YQ2023E034].

**Institutional Review Board Statement:** Not applicable.

**Informed Consent Statement:** Not applicable.

**Data Availability Statement:** The data presented in this study are available on request from the corresponding author. The data are not publicly available due to privacy.

**Conflicts of Interest:** The authors declare no conflicts of interest.

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
