# Peer review of "Analysis of Diffusion Characteristics and Influencing Factors of Particulate Matter in Ship Exhaust Plume in Arctic Environment Based on CFD"

_atmosphere, doi:10.3390/atmos15050580_

Round 1

Reviewer 1 Report (Previous Reviewer 1)

Comments and Suggestions for Authors

The authors generally satisfy the comments of the referee, I recommend the publication of the manuscript.

Comments on the Quality of English Language

The English of the manuscript is satisfactory.

Reviewer 2 Report (Previous Reviewer 2)

Comments and Suggestions for Authors all comments and questions have been answered, and in my opinion, the acceptance of the article is unimpeded due to the corrections made.

Reviewer 3 Report (Previous Reviewer 3)

Comments and Suggestions for Authors

All items mentioned in the previous review are answered and revised in the paper. 

This manuscript is a resubmission of an earlier submission. The following is a list of the peer review reports and author responses from that submission.

Round 1

Reviewer 1 Report

Comments and Suggestions for Authors

- The manuscript is not complete and its English is poor.

- The originality of the study is poor.

- Solution methods and procedures are not presented adequately.

- On page 7, the grid size is used instead of cell size in some sentences.

- What is total export/import flow?

- Boundary conditions and assumptions are not presented adequately.

- There are statements not supported by references or results, see, for example, first sentence of  Section 2.5.

- How the particle capture rate is defined. It is not clear.

- What is the resultant wind speed? Explain.

- Define the chimney angle and yaw angle using a figure.

- Same names should be used for variables throughout the manuscript.

- In Figure 7, x-axis should be defined.

- On page 5, in the second paragraph, the sentence ".......... exposed to the water is considered." should be corrected as "... exposed to the air is considered."

Comments on the Quality of English Language

English of the manuscript is poor and It needs improvement. There are wrong sentences, see, for example, on page 2 in the first paragraph, the sentence "As a global climate driver second only to CO2 ........". On page 10, the last sentence should be improved.

Reviewer 2 Report

Comments and Suggestions for Authors

In this article, the authors analyzed the emission characteristics of the exhaust columns of a commercial ship using CFD simulation and investigated the effect of the three factors of wind speed, yaw angle and chimney angle on particle emission.

1)      The first question is Does the examination of other pollutants such as NOx, HC and … have no effect on the melting of polar ice, and only particulate matter are important?

2)      Considering the impact of these particulate matter, can you suggest an alternative method to reduce it?

3)      What is the effect of particulate matter on polar ice?

4)      In this research, how can separation be obtained based on the size of particulate matter?

Also, consider the following comments to improve the article:

Line 1: State the CFD Simulation in the title.

Line 7-18: Quantitative results should be included in the abstract

Line 19: Sort keywords alphabetically.

Line 70-90: The literature review is limited. Has there been no research in this field?

Line 137-139: “In this paper, two cases with velocity ratio (resultant wind speed/exhaust speed) K of 0.407 and 0.815 were selected for comparison. “ What was the reason for choosing these numbers?

Line 349: The conclusion should be rewritten and the important results should be bolded.

Reviewer 3 Report

Comments and Suggestions for Authors

In this paper, authors describe about the diffusion research of ship exhaust particulate matter under low temperature environment. They clarified the diffusion law of ship exhaust particulate matter and the influence of three factors, namely, resultant wind speed, yaw angle and chimney angle, on the diffusion of particulate matter.

The researchers and engineers in the field are interested in these results.  The paper is well-organized. 

Items to be considered 1. It might be better that authors describes about how particulate matter affect the polar ice. 2. Could you please describe what the horizontal axis of the graphs shown in Figure.7 is. 3. If possible, could you please describe what kind of method should be considered to decrease the affect of the particulate matter on the polar ice?